# OpenReview forum: "Daunce: Data Attribution through Uncertainty Estimation"
_ICLR.cc/2026/Conference — Submitted to ICLR 2026_

### Official Review · Reviewer_2cRW · 2025-10-30

**Soundness:** 2
**Presentation:** 2
**Contribution:** 3
**Rating:** 4
**Confidence:** 3

**Summary:**

The paper proposes a data-attribution method based on uncertainty estimation. It perturbs the model to generate uncertainty estimates and computes attribution scores via uncertainty-aware covariance. The approach is claimed to scale to large models, including black-box LLMs. Experiments on vision and language benchmarks evaluate attribution quality and efficiency.

**Strengths:**

1. Introduces a novel data-attribution algorithm leveraging uncertainty estimation rather than relying on gradients or Hessians.
2. Extends to large-scale settings, even black-box LLMs, demonstrating good generalization beyond standard gradient-based attribution setups.

**Weaknesses:**

1. The method incurs additional $O(K·|D^k|)$ training cost and requires storing K models. Computational and memory trade-offs are insufficiently analyzed and not compared against baselines. The scale and choice of $|D^k|$ are not specified or discussed, making it hard to assess practical overhead.
2. Experimental settings and baseline selections are inconsistent. Missing feasible baselines and inconsistent settings weaken the strength of the empirical evidence.

**Questions:**

1. Readability issues and omitted explanations reduce clarity.

    a. Lines 150–152: explain the optimization motivation. Is the optimization intentionally structured to mimic the final form of influence-function objectives?

    b. Lines 179–180: clarify whether the second optimization approximates the Jacobian via logits.

    c. Lines 199–204: “Then” does not seem appropriate here; the derivation follows from Eq. (6) rather than the previous statement. Also, if Appendix E.2 shall be cited here?

2. Experimental design inconsistencies.

    a. Sec. 4.1: with CIFAR-10/ResNet-9, influence function with Lissa approximation should be feasible. Why not include this baseline?

    b. Lines 262–264: It would be better to explain why unit-normalized gradients imply replacing covariance with correlation, and formally define the correlation measure used.

    c. Sec. 4.2:  Why is EKFAC not compared, given its demonstrated applicability to LLMs?

    d. Lines 403–405: Why is the first-order term considered small when its magnitude appears comparable to the empirical-risk term for LLMs?

---

### Official Review · Reviewer_nktV · 2025-11-03

**Soundness:** 3
**Presentation:** 3
**Contribution:** 2
**Rating:** 2
**Confidence:** 4

**Summary:**

This paper presents an efficient and scalable data attribution approach that estimates the pairwise influence of each training sample on each test sample using a perturbation-based strategy. The method perturbs the target model to create slightly varied model instances and computes the covariance of per-example losses across these perturbed models as the attribution score. The proposed approach is evaluated in both white-box and black-box settings, showcasing its plug-and-play adaptability and generalisation across diverse domains.

**Strengths:**

**1. Well-motivated and clearly presented** This paper is well-motivated with proper background discussion apt summary of their proposed methods.

**2. Black-box and white-box applicability** The proposed method is applicable both in black-box and white-box settings which is of immense practical use in real life application of AI

**Weaknesses:**

**1. Limited novelty compared to TARK.** This paper criticises TRAK [1] in their motivation due to their projection error. However, their proposed primary equation (5) is directly related to Eqn (11) of TRAK where they both applied a uniformly distributed random matrix to approximate the computationally expensive Taylor expression. Ensembling from randomness in the approximation is also originally proposed in TARK. Therefore, the claimed novelty of this paper's in bringing scalability to TDA methods is limited.

**2. Unclear Eqn (11).** The approximation step of Eqn (11) is not clear, and no proof is provided in the main paper or in the appendix.

[1] Sung Min Park, Kristian Georgiev, Andrew Ilyas, Guillaume Leclerc, and Aleksander Madry. Trak: Attributing model behavior at scale. arXiv preprint arXiv:2303.14186, 2023

**Questions:**

It seems the paper's main contribution lies in adapting TRAK's data attribution method in black-box setting, especially applying in LLMs. Therefore, the technical novelty of the paper is limited.

---

### Official Review · Reviewer_t3qJ · 2025-11-10

**Soundness:** 3
**Presentation:** 3
**Contribution:** 2
**Rating:** 4
**Confidence:** 3

**Summary:**

This paper introduces DAUNCE, a new approach to TDA. The core idea of the paper is to replace the expensive second-order information computation (like the Hessian) by fine-tuning a collection of perturbed models and computing the covariance of the per-example losses across the models. By avoiding this expensive second-order information, the method tries to solve a major bottleneck for large models.

**Strengths:**

- As mentioned in the summary, the paper's primary contribution, DAUNCE, introduces a simple yet novel approach to TDA. Instead of relying on computationally expensive second-order information like the Hessian matrix, which is a major bottleneck for evaluating large models, it uses uncertainty estimation. The method of fine-tuning K perturbed models and calculating the covariance of their losses is an efficient and smart approach to avoid the bottleneck.
- The paper also conducted extensive experiments for different sizes of models, which proved that the method itself can be scaled up to LLMs (which was not possible before). The author also provided both white-box and black-box scenarios, which are really impressive since there were very few papers considering the black-box case.

**Weaknesses:**

- The computation cost would be eye-watering considering all the perturbed models the method used. From `figure 1 (a)` and `figure 6` in the paper, it seems like DAUNCE only outperforms other methods if `K` is at least `100`. Even with a LoRA rank of `64`, this still looks really expensive. The TRAK (as baseline) only needs one forward and backward pass, but DAUNCE needs `k` fine-tuning runs.
- It seems like the method saturates after `k` approaches 200 (from `figure 6`); it would be nice to see an analysis about when to stop adding the perturbed models. Otherwise, I would recommend adding some comments about it.
- `Line 406` claims "gradients are naturally small." It would be nice to clarify how small the gradient is by logging some results. Otherwise, this sounds arbitrary.

**Questions:**

- My main concern would be the computation costs, to my understanding this method needs to store `k` perturbed models, which is really expensive and not available for many people in the community, can you clarify the training time cost and storage cost?  If the training time and storage cost are significantly higher than the baseline, the improvement would be very small considering this much larger cost.
- The method itself looks like `Bootstrap aggregating with a covariance metric`, if this is the case, shouldn't a baseline with simple ensemble of `k` models be considered as well?

---

### Meta-Review · Area_Chair_ibQ8 · 2026-01-05

**Summary:**

This paper has been assessed by three knowledgeable reviewers who recommended its rejection (one straight and two marginal reject scores). The reviewers appreciated the presented approach to training data attribution using uncertainty estimation instead of Hessians, potentially enabling scalability to large models and even black-box LLMs. They found it well-motivated, clearly presented, and supported by substantial experiments. However, the reviewers were concerned with the method’s computational costs, requiring hundreds of perturbed models, and lack of clear guidance on trade-offs or stopping criteria. Technical novelty was considered somewhat limited compared to prior work, with unclear approximations and missing proofs, and empirical evidence is weakened by inconsistent baselines and omission of recent, strong methods. The authors did not provide a rebuttal.

**Reviewer Concerns:**

There were no rebuttal.

**Reviewer Scores:**

Without rebuttal, that would be highly unlikely.

---

### Decision · Program_Chairs · 2026-01-26

Reject